# Evidence for the Growth Superiority and Delayed Ovarian Development in Tetraploid Loach *Misgurnus anguillicaudatus*

Xiaoyun Zhou [1], Zexia Gao [1], Shuangshuang Luo [1], Junxiao Su [1] and Shaokui Yi [2,*]

1    College of Fisheries, Huazhong Agricultural University, Wuhan 430070, China
2    College of Life Sciences, Huzhou University, Huzhou 313000, China
*    Correspondence: 02844@zjhu.edu.cn

**Abstract:** The diploids and tetraploids of cyprinid loach *Misgurnus anguillicaudatus* coexist in many natural habits. The tetraploids generally exhibit superior growth performance and delayed gonadal development compared with diploids. To investigate the regulation mechanism of growth superiority and ovarian development in tetraploids, we first conducted a long-term culture experiment and histological observation. The tetraploids exhibited a higher growth performance and delayed ovarian development. Meanwhile, at a genetic level, an average of 6891 differentially expressed genes (DEGs) between diploids and tetraploids were identified from the brain, pituitary, liver, gonad and muscle using the RNA-seq method. Among these DEGs, as expected, some candidate genes, including IGF family genes, *somatostatin*, *leptin*, *cyp19a1b*, *gthα*, *lhβ* and *fshβ*, were detected, which play critical roles in the regulation of growth and gonad development of fish. In particular, the genes related to GH/IGF axis and growth factors, signal transduction, gonadal hormone and appetite were significantly increased in tetraploids. The clustering analyses of the key candidate genes showed that most key genes were up-regulated in the pituitary and gonad of tetraploids instead of other tissues. The dynamics of these key genes provide valuable genetic evidence for clarifying the growth superiority and delayed gonadal development of tetraploids. Moreover, this study also provides some clues for unveiling the genetic superiority of polyploidy species in other phenotypes.

**Keywords:** diploid; tetraploid; growth superiority; gonadal development; RNA-seq

## 1. Introduction

Polyploidy in fish is often associated with traits including larger body size, faster growth rate, improved longevity and better ecological adaptability compared to diploids [1]. However, even though polyploids typically have larger cells because of the increased amount of genetic material, most artificially produced polyploid fishes do not exhibit any growth advantage over their diploid relatives [2,3]. Some studies proposed that most animal polyploids have a decreased numbers of cells compared to the expectation from specific ploidy and maintain a similar body size to their diploid progenitors [3,4]. Alternatively, the polyploidy induction procedures, such as cold, pressure or another type of shock, may have negative effects on survival and growth [5–7]. In contrast to artificial polyploids, some natural polyploid fishes have larger body sizes than diploids. For example, polyploids in the genus *Barbus* [8,9] and the Japanese spined loach *Cobitis biwae* [10,11] are obviously larger than diploids.

The cobitid loach *Misgurnus anguillicaudatus*, also known as oriental weatherfish, is a small freshwater teleost which is widely distributed in eastern Asia. *M. anguillicaudatus*, for a long time, was employed as traditional Chinese medicine for the treatment of hepatitis, osteomyeitis, carbuncles, inflammations and cancers, as well as for patients' recovery from debilities caused by various pathogens and aging [12]. Due to its superior medicine value and excellent flavor, it has been widely consumed and favored for hundred years in East

Asia region. In the past few decades, the commercial culture of *M. anguillicaudatus* occupies a significant position in freshwater aquaculture production in Asia [13].

Aside from its significant medical and edible value, *M. anguillicaudatus* is full of interest to science because of the extensive karyotype polymorphism and its related atypical reproduction. Among the Japanese populations, most *M. anguillicaudatus* individuals are diploid, with 2n = 50 chromosomes, and undergo bisexual reproduction. Meanwhile, asexual diploid clones and natural triploids (3n = 75) are also observed in certain localities [14–16]. Among Chinese *M. anguillicaudatus* populations, in addition to the most common bisexual diploids, large numbers of tetraploids (4n = 100) have been recorded in the wild along the Yangtze River basin, central China [17–20]. These tetraploids reproduce bisexually, and are considered as autotetraploid, which may have arisen by the doubling of the entire genome of an ancestral diploid, based on cytogenetic results from FISH (fluorescence in situ hybridization) karyotypes and meiotic configurations [21,22]. Thus, *M. anguillicaudatus* comprises a diploid-tetraploid complex in central China [21] and is an extremely ideal fish model for studying genetic and phenotype consequences caused by ploidy variations in fish.

Growth is a primary economic trait for aquaculture species and severely affects the productivity and profitability of fish production. Using samples collected from a ploidy-complex population in the Yangtze River basin in central China, Feng et al. [23] demonstrated that tetraploid *M. anguillicaudatus* naturally had superior growth performance and achieved larger body sizes compared to diploids. Similar findings were also reported in Zhou et al. [24], who observed an improved growth rate of tetraploids compared with diploids and triploids under the same rearing conditions. In addition, Li et al. [25] reported a higher condition factor ($K$) in tetraploids than in diploids and triploids in wild populations. Despite the significant growth superiority in tetraploid *M. anguillicaudatus*, having been recognized in previous studies, the potential genetic mechanism underlying this predominant characteristic is not clear due to the lack of genomic and transcriptomic data.

Fish growth is primarily regulated by the hypothalamus-pituitary-liver (HPL) growth axis anatomically and the growth hormone/insulin-like growth factor (*GH/IGF*) axis of the neuroendocrine system [26,27]. In this axis, growth hormone (GH) is a pituitary hormone that participates in numerous physiological processes, including somatic growth, energy mobilization, gonadal development, osmoregulation and feeding behavior [28]. These physiological actions are triggered by the high-specificity binding between GH and its receptors (GHRs) in target tissues, such as liver, which is considered the main target. Except for the liver, muscle is also GHRs target, where GH also shows significant effects [29]. After GH binds to GHRs, it activates a post-receptor signaling system that stimulates the transcription of several target genes, including IGF-I and IGF-II [30]. As potent mitogenic hormone that induces growth and differentiation, IGFs mediates the growth-promoting capability of GH in a variety of target organs [31,32].

In the previous studies [23,24], we found that tetraploid *M. anguillicaudatus* had superior growth performance but the in-depth genetic evidence is absent. Therefore, in the present study, to uncover the genetic mechanism of growth superiority and ovarian development in tetraploids, RNA-seq was performed in both diploids and tetraploids with brain, pituitary, liver and muscle of the "growth axis", as well as the ovaries. This study will contribute to deciphering the molecular basis of growth superiority and the delayed ovarian development of tetraploid *M. anguillicaudatus*, as well as providing an ideal animal model for studying the balance between growth and reproduction.

## 2. Materials and Methods

### 2.1. Ethics Statement

All procedures involving fish were conducted in accordance with the Guide for the Care and Use of Laboratory Animals Monitoring Committee of Hubei Province, China, and the protocols were approved by the Experimental Animal Ethics Committee of Huazhong Agricultural University (permit number: HZAUFI-2018-026; 2018-3-27).

## 2.2. Fish Materials and Sampling

Parental diploid (4 female and 3 male) and tetraploid (3 female and 2 male) *M. anguillicaudatus* were obtained from Baishazhou Fish Market, Wuhan, China. The ploidy level of each individual was determined by a flow cytometer (Becton Dickinson Biosciences, San Jose, CA, USA) with DAPI staining according to the erythrocyte DNA content [33]. After artificial propagation in May 2019, diploid and tetraploid offspring were separately reared in an indoor pond under the same conditions at Fisheries Experimental Station of Huazhong Agricultural University.

After a one-year culture, 9 female diploids and tetraploids were randomly selected for RNA-seq, respectively. After anesthetizing with MS-222 (Sigma, St. Louis, MO, USA) at 100 mg/L, tissues including the brain, pituitary, liver, ovary and muscle were collected and immediately flash-frozen in liquid nitrogen.

## 2.3. Gonadal-Somatic Index and Gonadal Histology

Body weights (*W*b) and gonad weights (*W*g) of the loach individuals were measured at 7 timepoints during the two-year culture, and the gonadosomatic index (GSI) was calculated using the following formula: $GSI = 100 \times Wg/Wb$. The gonads were dissected, immersed in Bouin's fixative, dehydrated, embedded in paraffin and cut into 8-μm sections with a microtome. The slides were stained with hematoxylin and eosin (H&E) and observed under a microscope (Zeiss, Axiovert 200) and recorded with a digital camera. The histological analysis of ovary samples was performed using the criteria of Domagała et al. [34].

## 2.4. RNA Isolation and Sequencing

The total RNA of each tissue was separately isolated using TRIzol Reagent (Invitrogen, Waltham, MA, USA). RNA integrity and concentration were evaluated using Agilent 2100 Bioanalyzer (Agilent Technologies, Santa Clara, CA, USA) and NanoDrop 2000 spectrophotometer (Thermo Scientific, Waltham, MA, USA), respectively. An equal amount of RNA from three diploids or tetraploids were firstly pooled as one sample, and 3 replicated samples for each tissue were used for the following experiments. A total of 30 libraries were constructed using total RNAs from brain, pituitary, liver, gonad and muscle tissues. The libraries were constructed using the NEBNext® Ultra RNA Library Prep Kit for Illumina (NEB, Ipswich, MA, USA). Libraries were sequenced on an Illumina HiSeq™ 4000 platform with 150 bp paired-end mode.

## 2.5. De Novo Assembly and Annotation

Raw reads were filtered with Trimmomatic (*v*0.30) [35] to remove reads containing adaptor, low-quality reads and reads containing Ns. The high-quality clean reads from diploids and tetraploids were de novo assembled using Trinity [36] with default parameters. Then the TIGR Gene Indices clustering tools (TGICL) (*v*2.1) [37] were used to cluster and remove redundant transcripts. To evaluate the completeness of the transcripts, the BUSCO *v*2 pipeline [38] was performed against the dataset of Actinopterygii_db10. Subsequently, BLASTx alignment (*E*-value $\leq 1.0 \times 10^{-5}$) was performed with five public databases, including NR (NCBI non-redundant protein sequences), NT (NCBI nucleotide sequences), Swiss-Prot, KEGG (Kyoto Encyclopedia of Genes and Genomes) and KOG (eukaryotic Orthology Groups), and the best alignments were used to decide sequence direction of the unigenes.

## 2.6. Identification of Differentially Expressed Genes

To identify the differentially expressed genes (DEGs), the clean reads were mapped to the assembled transcripts using Bowtie2 [39]. The mapped reads were counted and then normalized to determine RPKM using RSEM [40]. Differential expression analysis between the different ploidy levels of each tissue was performed using DESeq2 [41]. The FDR (false discovery rate) method was used to determine the significance of gene expression difference. Unigenes with a *q* value < 0.05 and $|\log_2 (\text{fold-change})| > 1$ were considered

as DEGs. The gene ontology (GO) enrichment analysis of the DEGs was implemented by the GOseq package based on Wallenius non-central hypergeometric distribution [42]. The significantly enrichment pathways were defined as $p < 0.05$ using KOBAS 2.0 [43]. The heatmap was drawn with R package "pheatmap".

### 2.7. Validation of RNA-Seq Data via qRT-PCR Assay

To confirm the reliability of the data obtained by RNA-seq, 24 genes associated with growth and ovarian development were selected for validation using qRT-PCR. RNA sample of each tissue were reverse transcribed to cDNA using PrimeScript^TM RT reagent Kit with gDNA Eraser (Takara, Dalian, China). The primers are listed in Table S1. The qRT-PCR was performed on a CFX96^TM Real-Time PCR Detection System (Bio-Rad, Irvine, CA, USA). Each qRT-PCR mixture contained 0.8 μL forward and reverse primers, 1 μL template, 10 μL $2 \times$ SYBR mix (TaKaRa, Dalian, China) and 7.4 μL ddH$_2$O. Three replicates were utilized for each sample, and the *ef1α* gene was used as reference. The reaction conditions were 95 °C for 10 s, followed by 40 cycles of 95 °C for 5 s, 58–60 °C for 20 s and 72 °C for 20 s. The relative expression was calculated using the $2^{-\Delta\Delta Ct}$ method.

### 2.8. Statistical Analysis

The obtained data of BW and BL are presented as mean $\pm$ standard deviation (S.D.). Differences between tetraploids and diploids were determined using a one-way analysis of variance (ANOVA), followed by the Bonferroni post hoc test. Difference was regarded as significant when $p < 0.05$. Correlations between RNA-Seq and qPCR data were assessed though multiple linear regression using the coefficients of determination ($R^2$) and $p$ values. All statistical analyses were performed using SPSS 16.0 (Chicago, IL, USA).

## 3. Results

### 3.1. Growth and Ovarian Development of Diploids and Tetraploids

The morphological difference (Figure 1A,D) and histological features (Figure 1B,C,E,F) of the ovary in diploids and tetraploids were investigated, respectively. Tetraploid individuals exhibit superior growth performance, especially for body weight (BWs), than diploids in all stages (Figure 1G,H). At 12 and 24 months old, the BWs of tetraploids were 1.3- and 2.63-fold higher than those of diploids, respectively.

The fluctuation of the mean GSI in diploids and tetraploids during the first and second breeding season is shown in Figure 1I. The values of mean GSI in tetraploids were very low in the first breeding season and were significantly lower than those in diploids. Notably, a significant increase of GSI was observed in tetraploids in the second breeding season. The GSI of tetraploids is about 1.5-fold higher compared with the 24-month-old diploids. Meanwhile, we investigate the ovarian development of diploid and tetraploid individuals, the HE staining of ovarian oocytes revealed that tetraploid individuals exhibited a lagging oocyte development compared with diploids. At 12 months old, tetraploid *M. anguillicaudatus* exhibited a stage II ovary containing perinucleolar oocytes (PO) and few cortical-alveolar oocytes (CAO), whereas diploid individuals displayed a stage IV ovary, which was characterized by the appearance of late/mature-vitellogenic oocytes (LVO) (Figure 1B,C). In the second breeding season (at 24 months), tetraploid *M. anguillicaudatus* reached first sexual maturity, with large amount of late/mature-vitellogenic oocytes (LVO) filling the ovary (Figure 1E,F). The above results indicate that the first sexual maturity of tetraploid *M. anguillicaudatus* is at least one breeding season later than that of diploid individuals.

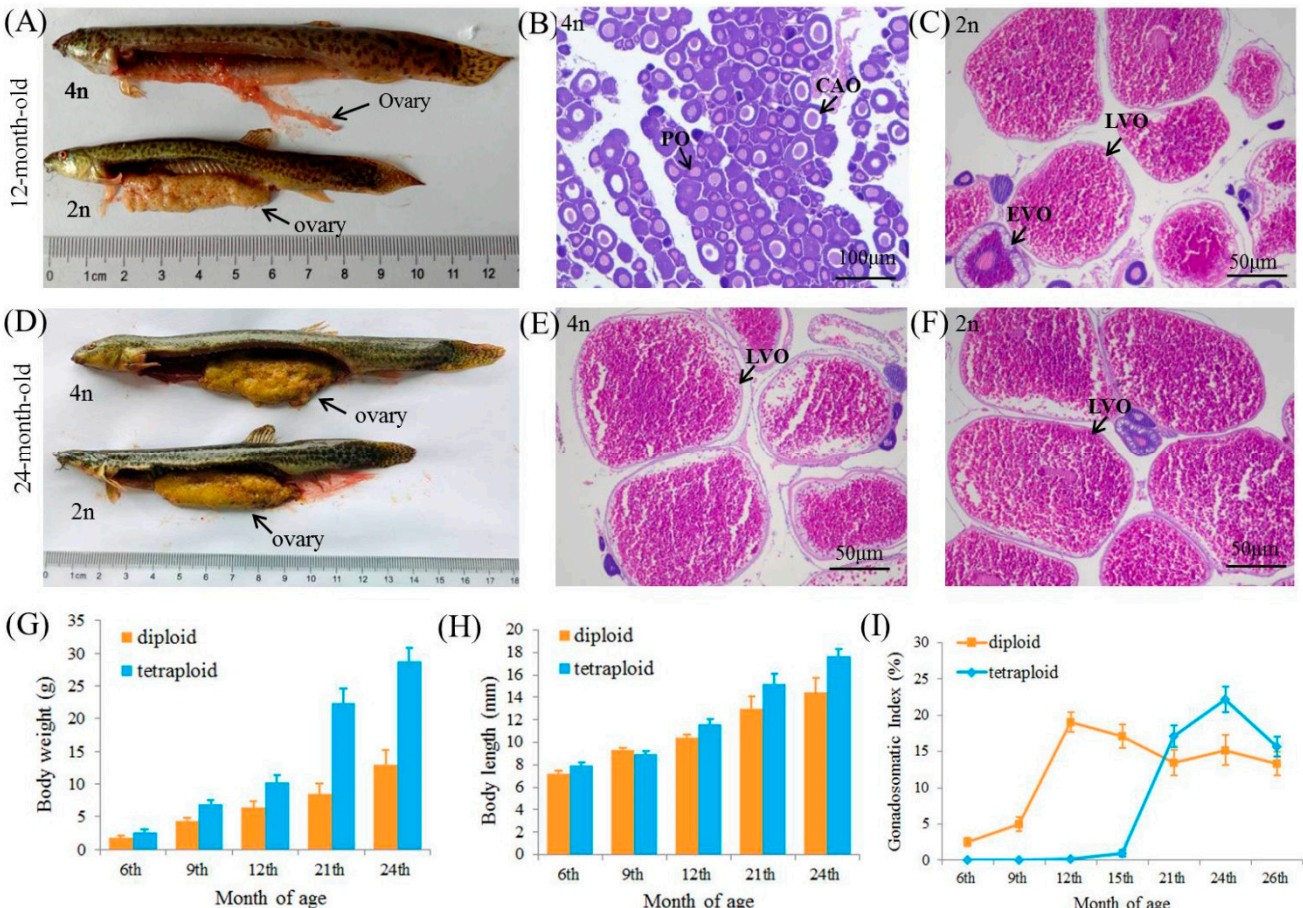

**Figure 1.** The ovarian development and body growth traits of tetraploid and diploid *M. anguillicaudatus*. (**A**,**D**) display ovarian appearance difference at 12 and 24 months old; (**B**,**C**,**E**,**F**) display ovarian histological difference at 12 and 24 months old; (**G**–**I**) display body weight, body length and gonadosomatic index difference between diploids and tetraploids in two-year culture experiments. PO, perinucleolar oocyte; CAO, cortical-alveolar oocyte, EVO, early vitellogenic oocyte; LVO, late/mature vitellogenic oocyte.

### 3.2. Assembly and Annotation of Transcripts

An average of 53,954,666 clean reads was obtained from each library. The sequencing information of this study is summarized in Table S2. A total of 253,559 transcripts with an average length of 938 bp and a N50 of 1823 bp were assembled using the Trinity. We then clustered the transcripts using the TGICL, which generated a total of 129,671 non-redundant unigenes with an average length of 620 bp and a N50 of 876 bp. The length distribution of unigenes was between 201 and 17,852 bp. The BUSCO analysis indicated that the completeness of the assembled transcripts was 94.1%.

To validate and annotate the assembled unigenes, the unigenes were subjected to BLASTx searches against five public databases. As a result, 46,832 (36.12%), 36,439 (28.10%), 27,607 (21.29%), 14,635 (11.29%) and 32,189 (24.82%) unigenes had homologous sequences in NR, Swiss-Prot, KEGG, KOG and Pfam, respectively (Figure S1). The annotation results for the NR database revealed the majority of the sequences identified with BLAST most closely matched genes from close-related fish species, as expected, such as *Cyprinus carpio* (20.08%), *Sinocyclocheilus rhinocerous* (18.15%), *Sinocyclocheilus anshuiensis* (15.36%) and *Danio rerio* (13.18%).

### 3.3. Identification of Differentially Expressed Genes

The differentially expressed genes (DEGs) between diploids and tetraploids in each tissue were identified by pairwise comparisons, and the number of DEGs is shown in Figure 2A. The details of DEGs are shown in Table S3. Notably, the number of DEGs identified among the gonad groups is obviously higher than that among the other groups. A total of 276 overlapped DEGs were obtained in all pairwise-comparisons (Figure 2B) and the GO enrichment analysis of these DEGs revealed that the metabolic process and catalytic activity were significantly enriched (Figure 2C). We found that some up-regulated genes in tetraploids are related to growth performance, such as *igf1*, *igf2*, *GHRHR*, *GRTP1*, *igfbp7*, *igfbp10* and *TGF-β*, while some reproduction-related genes, including *lhβ*, *fshβ*, *prl*, *slα*, *erα* and *cyp19a1b*, are down-regulated in tetraploids (Table 1). In the brain and pituitary, some pathways related to signal transduction, such as Notch signaling pathway, TGF-β signaling pathway and MAPK signaling pathway, were significantly enriched with the identified DEGs. Unlike the brain and pituitary, many pathways associated with lipid metabolism, amino acid metabolism and carbohydrate metabolism, such as tryptophan metabolism, glycerolipid metabolism, pyruvate metabolism, sphingolipid metabolism and ascorbate and aldarate metabolism, were significantly enriched with the DEGs identified in the liver. The enriched pathways in different tissues showed an obvious pattern, which was closely related to the tissue functions. In muscle, some pathways around endocrine system were significantly enriched, including the insulin signaling pathway, thyroid hormone signaling pathway and PPAR signaling pathway. These pathways play important roles in the growth regulation of aquatic animals.

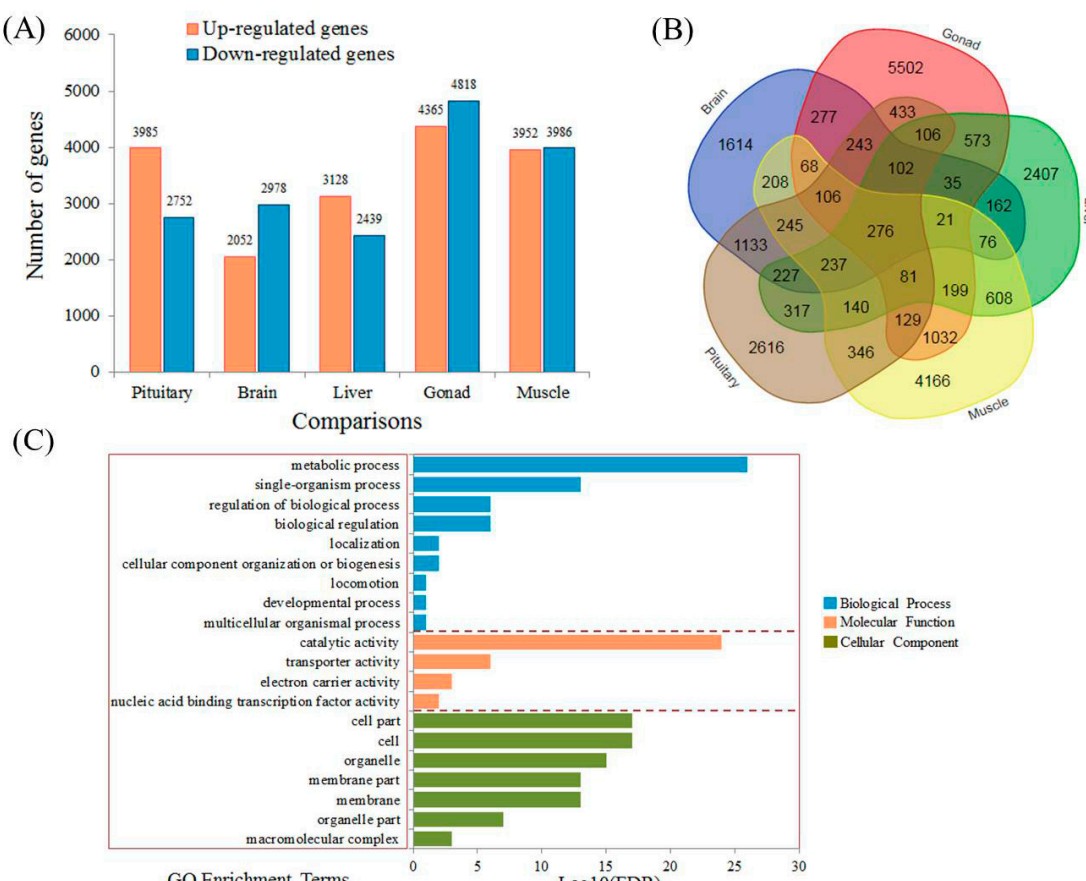

**Figure 2.** Differentially expressed genes (DEGs) obtained by pairwise compassion in five tissues of *M. anguillicaudatus*. (**A**) The number of identified DEGs in each tissue; (**B**) the five-way Venn diagram of the DEGs identified from the five tissues; (**C**) DEGs overlapped among tissues.

**Table 1.** Some key DEGs associated with growth and ovarian maturity identified by the pairwise-comparisons of diploid and tetraploid *M. anguillicaudatus*.

| Unigene ID | Log2FC | Annotation | Gene Name | FDR |
|---|---|---|---|---|
| TRINITY_DN27311_c2_g2 | 2.2012 | insulin-like growth factor I | *igf1* | $3.50 \times 10^{-10}$ |
| TRINITY_DN30348_c3_g1 | 1.7627 | insulin-like growth factor II | *Igf2* | $2.53 \times 10^{-10}$ |
| TRINITY_DN26313_c4_g1 | 2.0406 | growth hormone-releasing hormone receptor | *GHRHR* | $5.10 \times 10^{-8}$ |
| TRINITY_DN30415_c0_g2 | 1.1232 | growth hormone regulated TBC protein 1 | *GRTP1* | $3.94 \times 10^{-16}$ |
| TRINITY_DN38250_c0_g3 | 1.4433 | insulin-like growth factor-binding protein 7 | *Igfbp7* | $1.07 \times 10^{-28}$ |
| TRINITY_DN24955_c0_g1 | 1.3327 | insulin-like growth factor-binding protein 10 | *Igfbp10* | $2.28 \times 10^{-12}$ |
| TRINITY_DN25644_c0_g1 | 2.7081 | elongation factor 1-$\alpha$ | *ef1α* | $8.63 \times 10^{-4}$ |
| TRINITY_DN40659_c0_g1 | 2.2222 | transforming growth factor beta 1 | *TGF-β1* | $9.20 \times 10^{-15}$ |
| TRINITY_DN31880_c1_g1 | 2.6882 | transforming growth factor beta-3 | *TGF-β3* | $3.45 \times 10^{-109}$ |
| TRINITY_DN48440_c2_g3 | −1.2520 | luteinizing hormone-β | *lhβ* | $1.10 \times 10^{-234}$ |
| TRINITY_DN20712_c3_g3 | −2.9352 | follicle-stimulating hormone-β | *fshβ* | 0 |
| TRINITY_DN43724_c0_g4 | −1.0389 | brain aromatase | *cyp19a1b* | $4.04 \times 10^{-19}$ |
| TRINITY_DN33263_c0_g1 | −2.2631 | prolactin | *prl* | 0 |
| TRINITY_DN40445_c1_g4 | −1.8500 | somatolactin alpha | *slα* | 0 |
| TRINITY_DN23324_c0_g2 | −4.4445 | leptin | *leptin* | $8.32 \times 10^{-36}$ |
| TRINITY_DN28846_c0_g1 | 3.2100 | gonadal aromatase | *cyp19a1a* | $3.64 \times 10^{-43}$ |
| TRINITY_DN48025_c3_g3 | 1.2916 | estrogen receptor α | *erα* | $3.90 \times 10^{-5}$ |

The results of GO and KEGG enrichment analyses are shown in Table S4. Most GO items were assigned into biological processes and cellular components. To identify the biological pathways that might be associated with the growth superiority of the tetraploid *M. anguillicaudatus*, a total of 303 pathways were enriched with 4205 DEGs, such as the pathway of fat digestion and absorption, glycolysis/gluconeogenesis, citrate cycle, fatty acid metabolism, PPAR signaling pathway, insulin signaling pathway and protein processing in endoplasmic reticulum.

### 3.4. Candidate Genes Mediating Delayed Ovarian Development of Tetraploids

Gonadal development and maturation of fish are primarily controlled by the endocrine system consisting of the hypothalamus–pituitary–gonad (HPG) axis. Some key DEGs related to Jak-STAT signaling pathway, cytokine–cytokine receptor interaction, ECM–receptor interaction and cell adhesion molecules were detected. Notably, the lower expression levels of *cyp19a1b* (brain aromatase) and *erβ1* (estrogen receptor beta1) in the brain were decreased in tetraploids. In the pituitaries, the major hormones of reproduction, i.e., *gthα*, *lhβ* and *fshβ*, which directly control many aspects of gonadal development and function, exhibited a decreased expression pattern in tetraploids. In addition, the secondary hormones of reproduction, such as *gh* and *prl* (prolactin), were also decreased in tetraploid individuals. In contrast, the reproduction-inhibiting genes, such as D(2)-like dopamine receptors and *gabar* (γ-aminobutyric acid receptor), were up-regulated in tetraploid individuals. In the gonad, for genes related to reproductive and gonadal development, *cystatin, cystatin-B* and *cyp19a1a* (gonadal aromatase) were up-regulated, whereas the *vtgR* (vitellogenin receptor) were down-regulated. It was proposed that gonad GH/IGF-I axis also plays important roles in regulating the reproduction in teleosts, we found a lower expression of *ghr* (growth hormone receptor), *igf1r* (Insulin-like growth factor 1 receptor), *igf2, igfBP2b* (IGF-binding protein 2b), *igfBP3* and *igf2BP1* (IGF2 mRNA-binding protein 1) in the tetraploid gonads. In addition to the HPG axis, we found that the hepatic expression levels of *erβ1* (estrogen receptor bata1), *vtg1* (vitellogenin 1), *vtg2* (vitellogenin 2), *vtgC* (vitellogenin C), *vtg4* (vitellogenin 4), *fabp2* (fatty acid desaturase 2), *fabp10* (fatty acid-binding protein 10) and *leptin* were significantly lower in tetraploids compared with diploids. The delayed ovarian development of tetraploid *M. anguillicaudatus* also illustrated from the perspective of vitellogenesis pathway, as shown in Figure 3.

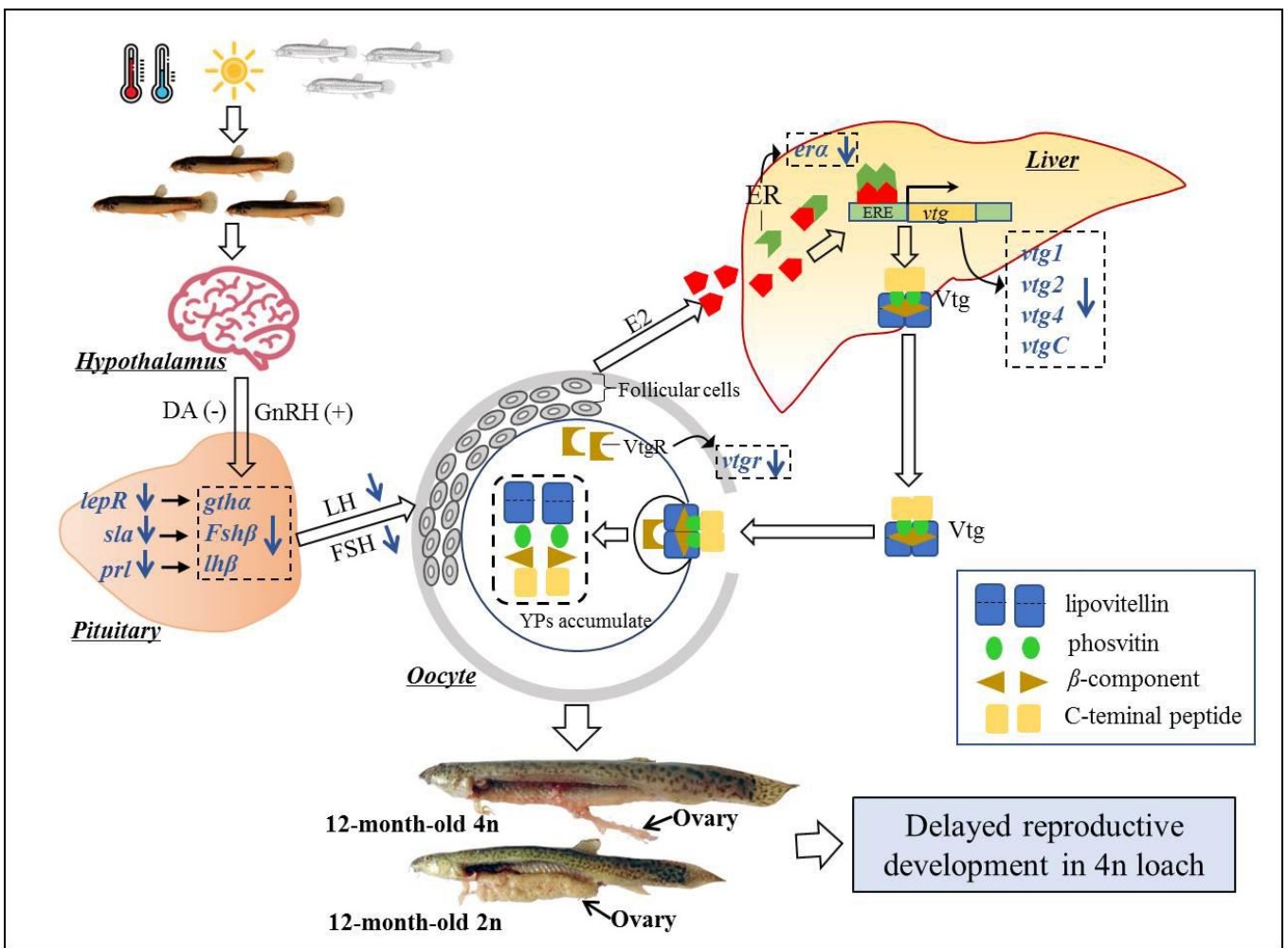

**Figure 3.** Illustration of the delayed ovarian development in tetraploid *M. anguillicaudatus* from the vitellogenesis pathway.

*3.5. Genetic Regulation Patterns of Growth Superiority in Tetraploids*

Since the GH/IGF axis plays a central role in the regulation of growth in fish, genes associated with GH/IGF axis are of significant importance in the enhancement of growth [44]. The up-regulated genes in tetraploid hypothalamus include the growth-related transcription factor *stat3* (signal transducer and activator of transcription 3) and *TGF-β* (transforming growth factor-beta). In the pituitaries, some key genes, such as *ghrhr* (growth hormone-releasing hormone receptor) and *igf1r* were significantly up-regulated, while *lepR* (leptin receptor) and tshR (thyrotropin receptor) were down-regulated in tetraploids. Importantly, hepatic *igf1*, as a primary mediator of growth-promoting effects of *gh* in fish, was up-regulated in tetraploids. Moreover, we found that the hepatic expression levels of *igfbp7* (insulin-like growth factor-binding protein 7), *igfbp10* and *grtp1* (growth hormone-regulated TBC protein 1) were also much higher in tetraploids rather than diploid while the down-regulation of *leptin* A2, *ss1A* (somatostatin-1A) and *ss2r* (somatostatin receptor type 2) were detected in tetraploids. The lower expression level of hepatic leptin and somatostatin in tetraploid could infer an improved appetite and increased fat content of the tetraploid loach.

We further analyzed the DEGs in muscle, which is the main target tissue of the GH/IGF system. In the present study, *RyR1* (ryanodine receptor 1) and *RyR3* were found to be up-regulated in tetraploids. The downstream $Ca^{2+}$-binding protein TnC (troponin C) was also up-regulated in tetraploids along with the other troponins, TnI and TnT. The

activation of $Ca^{2+}$ signaling as well as the abundance of the Tn complex may lead to severe muscle movement, which facilitated the increase in the size of skeletal muscle [45].

Given that the protein degradation through the ubiquitin–proteosome pathway plays a negative role in muscle growth. The expression levels of PI3K/Akt and ubiquitin-proteosome pathway genes in diploid and tetraploid were focused on, and we found that the expression level of Akt (serine/threonine-protein kinase), a critical signaling node of the PI3K/Akt pathway, was increased in tetraploid loach. As expected, a reduced expression of E3 ubiquitin–protein ligase, including *MuRF-2* (Muscle-specific RING finger protein 2), *Fbxo5* (F-box only protein 5), *Fbxo34* and *Fbxo43*, in tetraploid loach was observed. The relative lower abundance of E3 ubiquitin–protein ligase weakens protein degradation, which may partially responsible for the better growth performance of tetraploids.

Interestingly, we also found some genes related to appetite were differentially expressed between diploids and tetraploids. The *leptin*, *POMC* and *CART*, which have appetite-suppressing effects, were significantly down-regulated in tetraploids, whereas *AgRP* and *NPY*, which promote appetite, were up-regulated (Table 2). It could be inferred that the regulation of appetite-related genes play important roles in determining the growth superiority of tetraploids.

**Table 2.** Some candidate DEGs associated with appetite identified by pairwise-comparisons of diploid and tetraploid loach.

| Unigene ID | Log2FC | Annotation | Gene Name | FDR |
|---|---|---|---|---|
| TRINITY_DN23324_c0_g2 | −4.4445 | leptin A2 | *leptin* | $8.32 \times 10^{-36}$ |
| TRINITY_DN46583_c1_g2 | −2.3537 | Leptin receptor | *lepR* | $6.96 \times 10^{-16}$ |
| TRINITY_DN44455_c0_g2 | −1.3513 | Proopiomelanocortin | *POMC* | 0 |
| TRINITY_DN26070_c2_g3 | −1.3450 | Cocaine- and amphetamine-regulated transcript protein | *CART* | $3.24 \times 10^{-9}$ |
| TRINITY_DN34546_c0_g4 | 1.2470 | agouti-related peptide | *AgRP* | $0.17 \times 10^{-3}$ |
| TRINITY_DN32690_c1_g1 | 3.5570 | Neuropeptide Y | *NPY* | $2.75 \times 10^{-36}$ |

*3.6. Correlation of Ovarian Development and Growth in Tetraploids and Diploids*

To comprehensively understand the genetic regulation pattern of the tetraploid and diploid loach, 388 key genes which were associated with growth and ovarian development of fish were screened out according to the critical pathways and gene functions, and the genes that obviously exhibited higher expression levels in diploids or tetraploids were focused on. Overall, most genes showed an obviously different expression patterns in pituitary and gonad based on the clustering analyses (Figure 4). Notably, the expression levels of these genes did not exhibit an obvious pattern in the brain. Nonetheless, in the pituitary, a large number of genes were up-regulated in tetraploids. In muscle, we found some genes coding heat shock protein and genes related to TGF-$\beta$ pathway were up-regulated in tetraploids. In gonad, some key genes, such as serine/threonine-protein kinase and estrogen receptor beta 2, were up-regulated in diploids instead of tetraploids.

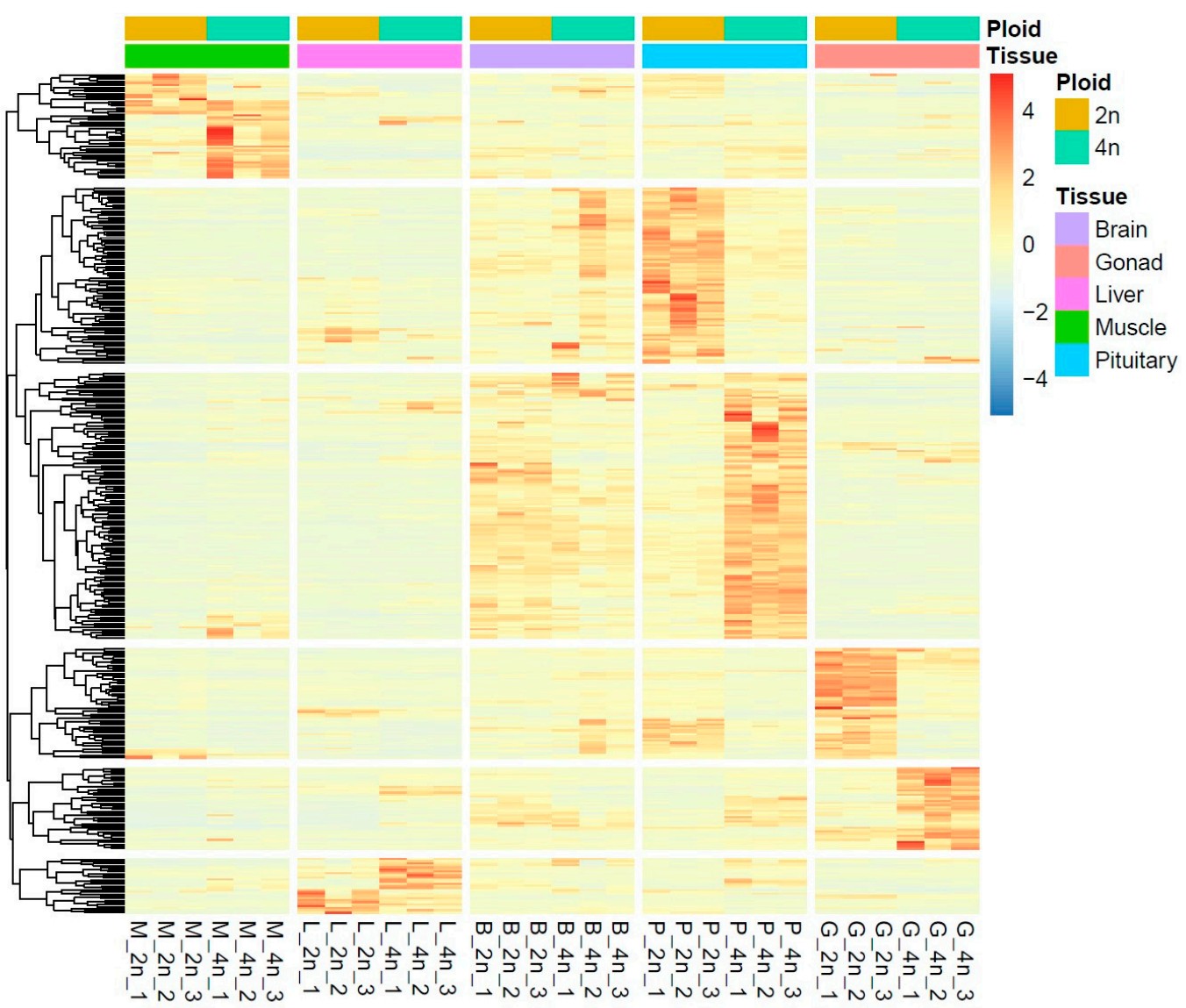

**Figure 4.** The heatmap of the key genes related to growth and gonadal development. The 2n and 4n indicate diploid and tetraploid individuals.

### 3.7. qRT-PCR Validation of DEGs

To validate the RNA-seq data, 24 important genes from the five tissues were selected for qRT-PCR analysis in this study. The results of qRT-PCR analysis basically coincided with the results of RNA-seq (Figure 5). The linear regression of $\log_2$ (fold-change) between qRT-PCR data and RNA-seq data were consistent with the correlation coefficient of 0.8114 ($p < 0.01$).

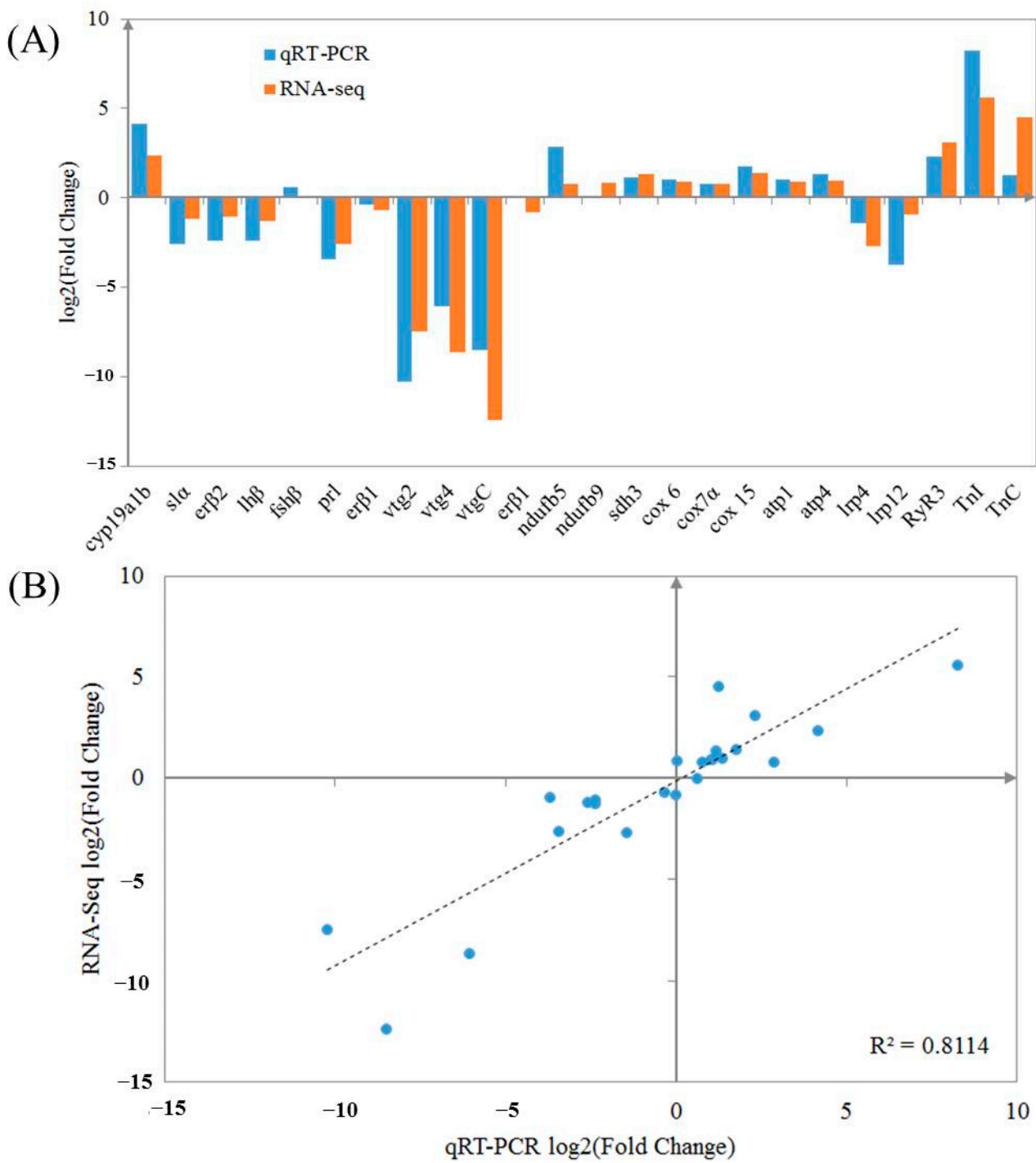

**Figure 5.** The expression tendency of 24 differentially expressed genes detected by RNA-Seq and qPCR.

## 4. Discussion

Polyploid fish has distinct advantages regarding some economics traits, such as rapid growth, extensive adaptability and disease resistance [24]. In this study, we found that the body length and weight of tetraploids were significantly larger than those of diploid loaches, which is consistent with a previous study [23]. To our knowledge, this is first study that reports the delayed ovarian development of tetraploids compared with diploids. Generally, in teleosts, growth and first sexual maturity are negatively corrective due to

the balance of energy metabolism. This could account for our findings in diploid and tetraploids. Gene expression analysis has been proposed as a powerful approach to detect the genesis basis of physiological and performance differences between fish with different phenotypes [46]. Thus, we used RNA-seq method to explore the molecular basis of the growth and ovarian development of the diploids and tetraploids.

*4.1. Up-Regulation of Some Key GH/IGF-Related Genes in Tetraploids*

The growth of vertebrates, including fish, is primarily regulated by the GH/IGF axis [47,48]. GH is a pituitary hormone regulating various physiological processes, such as somatic growth, immune function, lipid and protein metabolism [49], and GH stimulates hepatic and circulating IGF-I levels in teleosts [50]. In this study, we defined a large number of DEGs from GH/IGF axis. However, the expression level of *GH* in the pituitary was significantly lower in fast-growing tetraploids instead of diploids. Notably, the action of GH on phenotypic growth is largely determined not by the serum level of the hormone itself but rather by the interaction with its receptor GHR [44].

Previous studies have demonstrated a direct role of the IGF system in the determination of skeletal muscle growth in fish [51,52]. Within the IGF system, the expression levels of *igf1*, *igf2*, *igf1r*, *igfbp2b*, *igfbp3* and *igf2bp3* were significantly higher than diploid loach, indicating that these genes in GH/IGF axis might play an important role in the growth superiority of tetraploid loach. Moreover, since muscle growth results from a positive balance between protein synthesis and degradation and animals exhibiting better feed efficiency and growth performance also display slower rates of protein degradation [53]; therefore, it is essential to focus on the pathways and genes related to the regulation of proteolysis. Skeletal muscle protein degradation occurs through several pathways, one of the most important is the ubiquitin–proteosome pathway. This process could be suppressed by means of the inhibition of FOXO transcription factors by the activation of the IGF-1/PI3K/Akt pathway, blocking the up-regulation of E3 ubiquitin–protein ligase MAFbx and MuRF1 [29]. In this study, we compared the expression levels of PI3K/Akt and ubiquitin–proteosome pathway genes in diploid and tetraploids, and we found that, in addition to the expression levels of *igf-1* and *Akt2*, a critical signaling node of the IGF-1/PI3K/Akt pathway, were also increased in the muscles of tetraploids. Meanwhile, the decreased expression levels of E3 ubiquitin-protein ligase, including *MuRF-2*, *Fbxo5*, *Fbxo34* and *Fbxo43* in tetraploid loach was observed in tetraploids. The lower abundance of ubiquitin ligase could result in the inhabitation of protein degradation, which might be partially contributed to the superior growth of tetraploids. Additionally, Myostatin (MSTN) is a member of the transforming growth factor-β superfamily (TGF-β) and plays an important negative regulator of muscle growth in vertebrates [54]. *Somatostatin-1A* and *MSTN* were down-regulated in tetraploids compared with diploids, which also provided genetic evidence for the growth superiority of tetraploids. Somatostatin is a cyclic polypeptide that inhibits the release of a variety of regulatory hormones, including growth hormone, insulin and thyrotropin [55].

In this study, tetraploids exhibited lower hepatic *leptin* expression level compared with diploid loach, which indicated an improved appetite and increased fat content, because leptin plays critical roles in the regulation of body weight by inhibiting food intake and stimulating fat metabolism, as demonstrated in goldfish and green sunfish [56,57]. On the other hand, *leptin* played a role in reproductive development in teleost fish. For instance, in sexual mature Alantin salmon, the expression level of *leptin* was higher than that in immature individuals [58]. In common carp, recombinant Leptin protein could directly stimulate pituitary *gthα*, *lhβ* and *fshβ* expression and ovarian germinal vesicle breakdown in vitro [59]. In our study, in addition to the lower hepatic *leptin* expression levels, decreased leptin receptors were also identified in the pituitary and ovaries of tetraploid loach, which is consistent with the previous reports that significantly reduced pituitary *gthα*, *lhβ* and *fshβ* in tetraploid individuals.

*4.2. Genetic Difference of Ovarian Development between Diploids and Tetraploids*

In cyprinid fish, GnRH3 is the main form stimulating luteinizing hormone (LH) release [60]. In this study, the expression of *gnrh3* did not exhibit significant difference between two diploid and tetraploids. However, RNA sequencing and qPCR results indicated that the expressions of several dopamine receptor subtypes were increased in tetraploids. This is a clear evidence that dopamine acting via the pituitary D(2)-like dopamine receptor, which is a critical inhibitor of gonadotropin synthesis and secretion in teleosts [60]. Although the expression of *drd2* was not significantly different in the pituitaries between diploids and tetraploids, *drd3* and *drd4*, members of the D(2)-like family, were significantly increased in tetraploids. The elevated expression in pituitary D(2)-like dopamine receptors may therefore result in an increased inhibition of *gthα*, *lhβ* and *fshβ* in tetraploids.

Moreover, the gonad GH/IGF-I axis genes have been demonstrated to be involved in reproduction and play important roles in promoting spermatogenesis and oocyte proliferation and maturation [61]. In loach, the lower expression levels of *gh*, *ghr*, *igf1r* and *igf2* were detected in tetraploids, which could account for the delayed gonadal development of tetraploids. Generally, most polyploid fish do not differ significantly in body size from their diploid relatives [4]. In this study, we documented that autotetraploid loach exhibited higher growth performance than diploid loach. Given the "gene dosage compensation" phenomenon in polyploid species, we investigated the gene expression levels of transcripts, and only 7.82% (6891) of DEGs were detected between the two ploidies. It could be inferred that gene expression was not highly altered after polyploidization in loach, which was observed in previous studies of fish species [13,62]. In actual fact, the growth and sexual development of fish is a complex biological process [63], and the genetic evidence underlying growth superiority of polyploid fish still need further clarification.

## 5. Conclusions

This study investigated the regulation mechanism of growth superiority and ovarian development in tetraploid loach. A long-term culture experiment and histological observation revealed that the tetraploids exhibited a higher growth performance and delayed ovarian development. At genetic level, a number of DEGs between diploids and tetraploids were detected from the brain, pituitary, liver, gonad and muscle by transcriptomic analysis. Some key genes associated with growth and gonadal development, e.g., IGF family genes, *somatostatin*, *leptin*, *cyp19a1b*, *gthα*, *lhβ* and *fshβ*, were detected. Particularly, the genes related to GH/IGF axis and growth factors, signal transduction, gonadal hormone and appetite were significantly increased in tetraploids. These findings provide a better understanding of a balance act of growth and reproduction in fish as well as the genetic evidence for growth superiority and delayed ovarian development in polyploid fishes.

**Supplementary Materials:** The following supporting information are available online at: https://www.mdpi.com/article/10.3390/fishes7060322/s1, Figure S1: The annotation results of the unigenes with five public protein databases, Figure S2: The GO and KEGG enrichment of the differentially expressed genes in brain, pituitary, liver, gonad and muscle between diploids and tetraploids, Table S1: The information of the primers for qRT-PCR, Table S2: The summary of sequencing data of the RNA-seq libraries. Table S3: The details of DEGs identified by pairwise comparisons. Table S4: The summary of GO and KEGG enrichment analyses with all DEGs.

**Author Contributions:** Conceptualization, X.Z., Z.G. and S.Y.; investigation, X.Z., S.L., J.S.; data curation, S.L., J.S.; writing, X.Z. and S.Y.; revision, X.Z. and S.Y. All authors have read and agreed to the published version of the manuscript.

**Funding:** This research was funded by the National Key Research and Development Program (grant no. 2018YFD0900205) and the National Natural Science Foundation of China (No. 31472267).

**Institutional Review Board Statement:** The animal study protocol was approved by the Experimental Animal Ethics Committee of Huazhong Agricultural University (permit number: HZAUFI-2018-026, 2018-3-27).

**Informed Consent Statement:** Not applicable.

**Data Availability Statement:** The unigene sequences assembled with Trinity and function annotation of the unigenes have been deposited into the FigShare repository with the DOI links https://doi.org/10.6084/m9.figshare.21515457.v1 and https://doi.org/10.6084/m9.figshare.21515721.v1, respectively.

**Conflicts of Interest:** The authors declare no competing interests.

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
