# Peer review of "Evidence for the Growth Superiority and Delayed Ovarian Development in Tetraploid Loach Misgurnus anguillicaudatus"

_fishes, doi:10.3390/fishes7060322_

Round 1

Reviewer 1 Report

Zhou et al. explored the difference of growth and ovarian development in loach species that exists natural diploids and tetraploids. This study obtained some meaningful results with the long-term culture and RNA-seq method, like some genes associated with appetite. It entirely fulfills the scope of journal and also provides some new insights for the future studies of polyploid fish. I think some minor questions should be addressed prior to acceptance.

1. The grammar of English writing should be checked and revised for whole manuscript.

2. Line 68, “diploid-tetraploid complex in central China” has been mentioned, but the citation is absent.

3. Line 115, the DNA contents of loach samples were measured with flow cytometry, but the fluorochrome were not clarified in the text.

4. How many parental loaches were used for artificial propagation? Were the offspring’s from same bred family?

5. Line 198, change the “1st and 2nd” to first and second.

6. The trinity was used to assemble the transcripts, but I noticed that the genome of loach has been reported. Why the authors did not use the reference genome?

7. Line 241, the sentence should be removed from this section.

8. Line 311-312; please move the statement to discussion section.

9. The uppercase or lowercase letters in the Figure should be uniform.

10. Format of some references needs to be modified following the journal’s requirement.

Author Response

The point-to-point response has been attached, please check it.

Reviewer 2 Report

Revision

Title: appropriate to the analyzed content

Abstract: contains all the necessary information

Introduction: the status of Misgurnus anguillicaudatus in China, its use in medicine and economy are discussed. It also describes the ploidy nature of this species and its consequences. Growth, fertility in M. anguillicaudatus diploids, triploids, teteraploids and the mechanisms that regulate these processes have been described. Due to the fact that the general mechanisms influencing these processes are known, the authors of the publication focused on in-depth research on the influence of individual genes on growth, fertility and appetite in tetraploids. Molecular analysis of growth regulation, disturbed ovarian development M. anguillicaudatus is an interesting model organism for studying the balance between growth and reproduction.

Research methodology, selection of methods, equipment, reagents, programs, proper statistics.

Results

Fig. 1

Photographic documentation provides information on the appearance of diploids and tetraploids (Fig. 1 a, d)

Histology: very good quality histological images of the gonads, correct description of the figures, (Fig. 1 b, c, e, f)

The graphs (Fig. 1 g, h, i) show well the differences in growth and maturation of gonads in di- and tetraploids.

Correct caption for Fig. 1. A bit chaotic caption for Fig. 1. No detailed description of the histological preparations and no explanation of the meaning of the letter symbols placed on the histological pictures (in the caption under Fig. 1). In the description of the photos of the histological preparations, Fig. 1b, it can be written that it is pre-vitellogenesis, and Fig. 1c, e, f is vitellogenesis (stages of gonad maturity Domagała et al. 2013).

Annual development cycle of gonads of Eurasian ruffe (Gymnocephalus cernuus L.) females from lower Odra River sections differing in the influence of cooling water / Józef DomagaÅ‚a, Lucyna Kirczuk, MaÅ‚gorzata Pilecka-Rapacz. // Journal of Freshwater Ecology. 2013, vol. 28 iss. 3, s.423-437 DOI: 10.1080/02705060.2013.777855

Molecular analysis - the authors used modern methods of analysis. The results and the performed molecular analyzes are presented clearly by means of graphics. Performing analyzes of gene expression from di- and tetraploid organisms, from several organs, using modern methods, with validation and statistical analysis, makes the article important in the analysis of tetraploids. This publication provides the basis for a modern, reliable approach to research on organisms with multiplied genetic material. This is important from a utility point of view in the use of polyploidy in breeding. These studies could contribute to making Misgurnus anguillicaudatus a model species.

Particularly interesting is the analysis of candidate genes as well as the pathways of regulation of the processes responsible for the course of oogenesis and fish growth. Figures 3 and 4 are particularly interesting, as they clearly illustrate the regulations, relationships between organs and show the different expression of genes between di- and tetraploids.

Discussion: the results of the research are discussed and compared with the results of research on polypoloids to date. The discussion is written in a concise and transparent manner. The most important regulations between genes, organs, influence on sexual development, growth are explained. Many of these processes are known and described. This article, enriched with in-depth molecular analysis, explains these processes in an accessible way. This article can be successfully used in didactics, in explaining the molecular mechanisms of growth and sexual maturation of fish with reference to di- and polyploidy.

Obviously, since gonad development and fish growth are complex processes, this research requires further analysis.
